

# Homogenous regions based on extremogram for regional frequency analysis of extreme skew storm surges

Marc Andreewsky[1], Samuel Griolet[2], Yasser Hamdi[3], Pietro Bernardara[4], Roberto Frau[1]

[1]Department EDF-R&D-LNHE, Chatou, 78401, France
[2]Polytech Lyon, Rochetaillée sur Saône, 69270, France
[3]IRSN, Fontenay-Aux-Roses, BP17, 92262, France
[4]EDF Energy R&D UK Center, SW1E5JL, UK

Correspondence to: Marc Andreewsky (*marc.andreewsky@edf.fr*)

**Abstract**. To resist marine submersion, coastal protection must be designed by taking into account the most accurate estimate of the return levels of extreme events, such as storm surges. However, because of the paucity of data, local statistical analyses often lead to poor frequency estimations. Regional Frequency Analysis (RFA) reduces the uncertainties

associated with these estimations, by extending the dataset from local (only available data at the target site) to regional (data at all the neighboring sites including the target site) and by assuming, at the scale of a region, a similar extremal behavior. RFA, based on the index flood method, assumes that, in a homogeneous region, observations at sites, normalized by a local index, follow the same probability distribution. In this work, the spatial extremogram approach is used to form a physically homogeneous region centered on the target site. The approach is applied on a database of extreme skew storm

surges and used to carry out a RFA.

**Key words:** Regional Frequency Analysis; Spatial-extremogram, Surges.

## 1 Introduction

To resist marine submersion, coastal protection must be designed by taking into account the most accurate estimate of the return levels of extreme events, such as extreme sea level or storm surges.

When performing a local analysis in order to estimate high return levels, the local duration of observation is often too low to be able to obtain precise results on the estimates of return levels that are seeking (associated typically with a return period of *100* or *1000* years). For example, storm surge records calculated from tidal gauge measurements from one site are usually shorter than *30* years.

These uncertainties can be reduced by a Regional Frequency Analysis (RFA) developed by Dalrymple (1960), which tries

to exploit the similarities between sites. This kind of analysis is based on the index flood method and assumes that within a homogeneous region, extreme events normalized by a local index representing local features, are drawn from a common regional distribution.

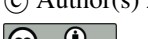



*Marc Andreewsky*

The principle proposed by Dalrymple (1960) and the approach developed by Hosking and Wallis (1997) were applied, for example, for skew storm surges (Bardet et al., 2011; Bernardara et al., 2011).

The grouping of sites into homogeneous regions defines the way to exploit regional information and, then, can have a significant impact on final results. Numerous hydrological papers have tried to address the formation of homogeneous

regions based, for example, on the study of the variables or parameters that describe at least in part the phenomenon of interest and which are physically related to it.

Authors such as Gabriele and Chiaravalloti (2013) recommended the use of meteorological information to delineate homogeneous regions and to carry out RFA of rainfall. However, a bibliographical review indicates that, before Weiss et al. (2013) and Weiss et al. (2014a), no specific method for marine hazards was developed to delineate homogeneous

regions via a physical approach. The criteria of merging sites in a homogeneous region were mainly based on statistical arguments, thereby excluding physical considerations. For example, by using data from 18 sites located on the west French coasts to extract storm surges, Bernardara et al. (2011) estimated extreme and the whole area was taken as homogeneous based on a statistical test of regional homogeneity.

In order to perform a RFA of extreme skew storm surges, Weiss et al. (2013) proposed a method based on physical

considerations to delineate and form homogeneous regions which depend on the extreme storms. The method, which has also been applied to waves (Weiss et al., 2014a), is based on typical storm footprints which are identified through a clustering algorithm which uses a criterion based on probabilities linked to storm propagation. Sites are then grouped into the different regions representing storm footprints. These regions can be considered as physically homogeneous because the same storms likely impact all sites inside a given region, and any storm on a region will likely remain enclosed in this

region.

However, the delineation of a homogenous region usually leads to the problem of the so-called "border effect". Indeed, if one is interested in a target site which is very close to the region limit, the information at the site located on the other side of the region is excluded, even though both sites offer similar information and have similar asymptotic properties. For example, in the physically homogeneous regions formed obtained by Weiss et al. (2013), it may be noted that, two French

sites, Boulogne-Sur-Mer and Calais are located in two physically different homogeneous regions while these cities are very close (they are only separated by about thirty kilometers). We can notice similar issues for all areas composed of two sites located at each side of a border between two regions. A physically homogeneous region defined by Weiss et al. (2013) is a typical storm footprint, and it can be expected, in fact, that very close sites facing similar storms are in the same area. Moreover, Weiss et al. (2013) gather in the same region very distant sites, which raises the question of whether there are

remaining traces of heterogeneity, even in a region considered statistically homogeneous. Acreman and Wiltshire (1987) have suggested that the sites located near the border between 2 regions could be considered partially owned by each of those two regions. However, Burn (1990) notes that there is no need to define boundaries between regions and a particular region can be defined for each site (which consists of sites similar to the site of interest in terms of extremes).

To form a physically homogeneous region centered on a target site, Hamdi et al. (2016) had recently proposed an approach

using the spatial extremal dependence between observations (the spatial extremogram) to measure the neighborhood





between sites. Herein, we define a pairwise distance between sites and we use the spatial extremogram approach for the RFA applied on extreme skew storm surges. The composition of regions built herein and which can be thought of as neighborhoods is based on the similarity of sites attributes. The higher the value of the spatial extremogram between the target site and another site is, the greater the dependency of extreme storm surges; therefore indicating that storms

impacting the target site tend to also impact the other site which can be included in the region of the target site. Indeed, in a specific region, the process generating storms and impacting the target site will also tend to impact the other sites in the region and vice versa. We can then consider that the processes generating storms in a region are physically homogeneous. So it is assumed that sites, with a sufficiently high value of the spatial extremogram with the target site, may be included in the same physically homogeneous region or, better, the region of influence of the target site. The region may also be

regarded as a typical storm footprint in the neighborhood of the target site.

Once a physically homogeneous region is formed around a target site, the statistical homogeneity is then checked. The whole procedure to estimate the regional law (and, in particular, the dependence model and the way to calculate the effective duration) is then applied in the same way as Weiss et al. (2014), starting from the physically homogenous regions defined from the spatial extremogram.

The detail of the methodology is described in section 2. In section 3, one will find an application of this method carried out for a database of extreme skew storm surges collected at 67 sites located along the Spanish, French and UK coasts. In order to compare the results obtained in this study with those of Weiss (2014c), the database used in section 3 is the same used by Weiss (2014c).

## 2 Methodology

The objective is to form physically homogeneous regions for RFA on extreme skew storm surges. The proposed method is based on the use of the spatial extremogram values.

### 2.1 Formation of physically homogeneous neighborhood of a target site by using the spatial extremogram

Let $X$ be the random variable representing the skew storm surges at a site *S1*, (the target site for instance), and $Y$ be the random variable for the skew storm surges at site *S2*. Let $q1$ and $q2$ be the thresholds above which skew storm surges are

considered as extreme.

If $h$ is a time offset, we denote $X_h$ the time series of $X$ shifted by $h$: $X_h(t) = X(t + h)$. For $h = 0$, we have the same series. We consider that the sites, *S1* and *S2*, are inside the same physically homogeneous region if at least a part of extreme skew storm surges from each site are likely to be simultaneously generated by the same storms (for the same time lag), which means that the extreme skew storm surges of the two sites are dependent. In the spatial and pairwise dependence description

we should also include the temporal dimension, because storm conditions can last several days. Then, a pairwise





*Marc Andreewsky*

probability of dependence can be defined by the spatial extremogram coefficient $\rho(X, Y, h)$ for bivariate time series, used also by Davis et al. (2011) and defined within Eq. (1):

$$\rho(X, Y, h) = \lim_{n \to \infty} ( P(q_{X,n}^{-1} \times X_h \in A | q_{Y,n}^{-1} \times Y_0 \in B) ). \tag{1}$$

Where $q_{X,n}$ and $q_{Y,n}$ are the $(1 - 1/n)$-quantiles of the distribution of $X$ and $Y$, and A and B are finite intervals that are bounded away from 0 ($A$ and $B$ will be included inside $[1; +\infty[$).

Davis et al. (2011) defined the natural estimator $\hat{\rho}(X, Y, h)$ of $\rho(X, Y, h)$, the empirical spatial extremogram, which will be used in this study, and which can be written as follows:

$$\hat{\rho}(X, Y, h) = \frac{\sum_{t=1}^{D-h} I_{[X(t) > q_X \text{ and } Y(t+h) > q_Y]}}{\sum_{t=1}^{N} I_{[X(t) > q_X]}}. \tag{2}$$

Where $D$ is the number of data occurring at the same time at the site S1 and S2, and where $q_X$ and $q_Y$ are extreme quantiles. Note that, we have to ensure that $D$ is large enough to have the guarantee that the probability of dependence is rather significant. Of course, we have $\hat{\rho}(X, Y, h) \in [0; 1]$ and $\hat{\rho}(X, Y, h) \neq 0$ indicates some dependency between $X$ and $Y$ on extreme values. A threshold $\rho_0$ is defined to indicate if the probability of dependence between $X$ and $Y$ is high enough to consider S1 and S2 inside the same physically homogeneous region: if $\hat{\rho}(X, Y, h) > \rho_0$ then *S1* and *S2* are considered to be part of the same physically homogeneous region. There is very often a residual probability of dependence which can be considered as noise, even between sites far enough; therefore, $\rho_0$ has to be great enough to make sure that the sites only linked by a residual probability of dependence (considered as noise) do not belong to the same region. The minimal value of $\rho_0$ can be found when analyzing the probability of dependence between all sites and the target site which allows for the evaluation of the order of magnitude of the maximum residual noise $m_r$. The determination of $\rho_0$ has to allow for the merging of any sites within the same region which have an extreme dependence with the target site, and to put aside sites only linked to the target site by a probability of dependence considered as residual noise.

The empirical quantiles $q_X$ and $q_Y$ are set in order to select in the $X$ and the $Y$ series only few storms per year, which allows for the computation of the empirical spatial extremogram from the biggest storm of each year. Moreover, $h$, the lag time, is large enough to allow a storm that occurred at one site to propagate eventually to the other site. If $h$ is set to several values then, between two sites, several values of $\hat{\rho}$ are estimated and the greatest value of $\hat{\rho}$ is kept.

Finally, the neighborhood of a target site is formed by all sites which satisfy a probability of dependence with the target site greater than $\rho_0$. From a statistical point of view, this means that if a storm affects a given target site, this storm will likely impact (and therefore not systematically) only sites enclosed in the target site's region and vice versa. The neighborhood of the target site can be seen also as the region of influence around the target site (the so-called RoI approach, Burn, 1990).



### 2.2 Independent storms extraction

In this section, we describe the procedure to construct a sample of independent storms. The procedure is the same as the one used in Weiss et al. (2013) and applies to each neighborhood of each target site obtained from section 2.1. The procedure is summarized below:

We define storm as a physical event that generates extreme skew storm surges in at least one site in the neighborhood of a target site (the study area). In this section, an observation is considered as extreme for a given site, if it exceeds $q_p$, where $q_p$ is the $p$-quantile of the initial at-site skew surge series, with $p$ close to $1$. Thus, a site is considered impacted by a storm if $q_p$ is exceeded. Moreover, at a given site, one or more extremes can occur during the same storm, according to its duration. When several extremes appear, only the maximum value is retained. This operation allows us to get independent

extremes extracted from storms at site scale.

The detection of storms which propagate both in space and in time, rely on a spatio-temporal declustering procedure. The main idea is that extreme neighbors in time and in space are considered to be part of the same storm. In other words, two extremes are spatio-temporal neighbors if they:

-    are among the $\gamma$-nearest neighbors of each other.

-    occurred within $\Delta$ hours,

Thus, to detect a storm, three parameters are needed: $p$, setting its impact on a given site, and $(\Delta, \gamma)$ which are linked and depend on its spatio-temporal propagation. In order to guarantee an accurate detection of the physical events, $(p, \Delta, \gamma)$ should be chosen correctly. Since we work on the same database of skew storm surges as the one used by Weiss et al. (2013), those parameters will be the same as their, who obtained them after various tests: $p = 0.995$, $\Delta = 24\ h$, $\eta = 14$. We

also, like in the study of Weiss et al. (2013), consider here the hypothesis often accepted in the literature that the declustering procedure leads to a sample of independent storms. In order to detect storms, Nissen et al. (2010) used a similar way as described above but from wind speed observations.

Therefore, we assume that this procedure allows for the construction of independent storms for each region defined as the neighborhood of a target site.

The storms extracted in this section represent physical events generating extremes; however, for statistical aspects, and in order to build the regional sample, a sub-selection of these storms is extracted in order to focus on the most intense events. In particular, like Weiss (2014c), we redefine storms in such a way that, on average and at each site, there is $\lambda$ storm(s) a year. Weiss (2014c) suggest to use $\lambda=1$, and we set $\lambda$ to the same value, which enables us to carry out the statistical analysis from the most the biggest storms. Bernardara et al. [2014] recommend this ''double-threshold'' procedure to address auto-

correlated environmental variables using the peak over the threshold method.



*Marc Andreewsky*

## 2.3 Regional statistical homogeneity

RFA of extreme storm surges requires statistically homogeneous regions. Like in Weiss et al. (2013), the physically homogeneous regions obtained with the method described in section 2.1 should be also statistically homogeneous. Two tests are used to verify that regions are statistically homogeneous:

The Hosking and Wallis test (Hosking and Wallis, 1997) can be used to assess the statistical homogeneity of a region. Their heterogeneity indicator $H$ measures whether the dispersion between sites is similar to a value that would be expected in a statistically homogeneous region. Hosking and Wallis suggest that a group of sites may be regarded as "acceptably homogeneous" if $H < 1$, "possibly heterogeneous" if $2 > H \geq 1$, and "definitely heterogeneous" if $H \geq 2$.

The discordancy criterion $D_c$ of (Hosking and Wallis, 1997) can identify discordant sites by indicating if, in a given region,

a site is significantly different, in terms of L-moments, from the other sites. If $D_c > 3$, a site can then be considered as discordant.

For each region defined as the neighborhood of the target site, the homogeneity and the discordance test are performed. If one site is found to be discordant, then a second neighborhood of the target site is defined without the discordant site. For target sites where the quantiles' estimations are particularly important, an RFA can be carried out on the two neighborhoods

(the one with the discordant site and the one without it) in order to compare the results and to keep the highest one (for conservative reason).

Finally, in all cases studied, we checked that the target site is not on the edge of its region. This must be the case most of the time. But, if it is not the case, we consider that another method (for example the one developed by Weiss, 2014c), could be better adapted to estimate the quantiles at the target site.

## 2.4 Fitting the regional samples, and quantiles estimations

RFA assumes that within a homogeneous region, extreme events normalized by a local index are drawn from a common regional distribution. The local index represents local specificities. Here again, the same procedure as the one developed by Weiss (2014c) is used. This procedure is summarized below:

For each site $i$, let's denote $u_i$ the storm threshold which is exceeded by the skew surges $\lambda$ times per year on average. Let

the ni-sample $X_i$ be the exceedance of $u_i$. It is assumed that $X_i$ is drawn from a GPD law.

The annual quantile $\mu_i$ of each site is used to normalize local sample $X_i$. $Z_i$ denotes the normalized local sample for the site $i$.

For each storm defined in section 2.2, we keep only the maximum normalized skew storm surge in the regional sample, ensuring sample independence. For statistical aspects, Weiss (2014c) use a threshold which allows to select the most

intense storms, and corresponds to a value of $\lambda$ equal to $1$. $M_s$ denotes this sub-selection of the maximum normalized skew storm surge sample for a specific region centered on a target site which corresponds to $\lambda = 1$.

A Kolmogorov-Smirnov test is carry out in order to check whether the law of $M_s$ can be considered as the law of $Z_i$. If it is the case, $M_s$ can be considered as the regional sample.



A Generalized Pareto Distribution (GPD) is fitted to the regional sample taking into account the seasonality (the threshold of the regional GPD law is equal 1, in the case of peak over the threshold method applied to the regional sample). Seasonal effects can be modelled through a sinusoid and the regional distribution is a discrete mixture of GPD where the scale parameter varies periodically and smoothly across the seasons of occurrence of storms. 4 seasons are considered here:

5 summer (June, July, August,), autumn (September, October, November), winter (December, January, February) and spring (March, April, May). The AIC criteria is taken into account to select the more adequate distribution. The good adequacy of the fit curve to the regional sample is also checked visually.

Regarding the effective duration of the regional sample, which is denoted $D_{eff}$, this value depends on the effective duration $d_i$ of each local sample (i varying from 1 to N, where N is the number of sites that are included in the region) and is closely

10 related to the spatial dependence (characterized by a $\varphi$ function). $D_{eff}$ is estimated by Weiss et al (2014b). Details about $\varphi$ and $D_{eff}$ are given in Appendix A.

Finally, the local $T$-return level $x_T^i$ of the target site i, is calculated within the following equation:

$$x_T^i = \mu_i y_T \tag{3}$$

Wherein $y_T$ is the regional $T$-return level of the region focused on the site *i*.

15 ### 3 Application

#### 3.1 Skew storm surge data

The database used in this study is the same as the study of Weiss, (2014c), which enables a simple comparison between our results and the results obtained by Weiss, (2014c). The raw data used is a temporal series of hourly sea level observations collected at 67 ports along the Spanish, French and U.K. coasts (see Fig. 1). In Appendix B, we recall some

20 elements about the database.

The construction of physically and statistically homogeneous regions within the results of Weiss (2014c) are presented in Fig. 2.

Note that some sites, like Calais in northern France (inside the red circle on Fig. 2), are located very close to the border between two regions. Nevertheless, on both sides of the border the process generating storms is likely the same. In addition,

25 the region 1 is very large (for example Boulogne on the extreme North of France is inside the same region as Saint-Jean-de-Luz at the extreme South of France, or in the north of Spain), with perhaps differences between the process generating storms in the north of France and in the north of Spain (which may cause traces of heterogeneities). Furthermore, in the study of Weiss (2014c), the region where La Rochelle and Brest are included has a heterogeneity measure equal to 1.1, and then is only considered as possibly statistically homogenous (according to the criteria of Hosking and Wallis described

30 in section 2.3).




*Marc Andreewsky*

### 3.2 Formation of homogeneous region centered on a target site

Quantiles $q_X$ and $q_Y$ should not be too small, otherwise the spatial extremogram won't be performed from extreme values. In addition, quantiles $q_X$ and $q_Y$ should not be too large, otherwise the spatial extremogram won't be performed on enough values. So there is a trade-off to be found.

5    Values of $q_X$ and $q_Y$ are tested in order to select first 4 and then 6 storms a year, which finally give information from the spatial extremogram that led to similar conclusions. Therefore, the empirical quantiles $q_X$ and $q_Y$ are set in order to select (in the *X* and the *Y* series) only 4 storms a year. This value allows for the computation of the empirical spatial extremogram from the biggest storm of each year.

Moreover, *h*, the lag time, has to be large enough to allow a storm which occurs at one site to propagate eventually to the 10    other site. If we note $d_s$, the time between two skew storms surges (it means about $\pm12h$), then tests performed with *h>24h* show little interest compared with tests where *h=0* or *h=ds*.

*h* is finally set to two values : *0* and $d_s$. Then, between two sites, two values of $\hat{\rho}$ are estimated and the greatest value is kept.

And at last, we set $\rho_0$ to *0.3*, which allows for the elimination of any sites associated with a value of the spatial extremogram 15    that looks like a residual noise from the target site's region.

However, we will see, that for some special cases (rare cases), one can consider including a site even if the spatial extremogram with the target site shows a slightly smaller correlation than $\rho_0$.

### 3.3 Application for several target sites

We apply our methodology to three sites for which we estimate the 1000-years return period quantiles.

20    ### 3.3.1 The physically and statistically homogeneous region for Calais

As previously noted, the particularity of the Calais site is that it is located close to a border of one of the regions found by Weiss (2014c) or Weiss et al. (2013). The application of the spatial extremogram (see Fig. 3) leads from Calais to the region illustrated in Fig. 4.

The Fig. 3 shows, on the vertical axis, the probabilities of the extremal dependence. The x-axis represents the sites which 25    are sorted in an ascending order (based on the geographical distance to the target site, the closest sites from the target site are on the left). The sites with extremal dependence probabilities greater than a threshold of *0.3* (represented by a red line) are considered as potential neighbors of the target site (Calais) and are thus part of the region of interest (of Calais) considered to be physically homogeneous (see Fig. 4). In brackets, next to the name of each site i, are indicated the durations for the year in which the tide gauge of the site i and the tide gauge of the target site have operated simultaneously; 30    therefore, it is the number of years on which the extremogram was calculated. For example, between Calais and Dunkerque it is *26* years. If this period is too small, the probability of the extremal dependence may not be relevant, and one can ask



whether it is appropriate to add the site to the region of the target site. This question must be answered case by case. The region built around Calais is shown in Fig. 4.

As shown in Fig. 4, the region around Calais is slightly smaller than the one obtained by Weiss (2014c) or Weiss et al. (2013), but more centered on Calais (which is no longer located at the border of a region). This region is considered as a

physically homogeneous region centered on Calais. We will see that, in general, we find smaller areas than those found by Weiss (2014c) or Weiss et al. (2013). But the advantage of a smaller region which is centered on a target site is that it is, most likely, more physically homogenous.

Once the physically homogeneous regions have been identified, the statistical homogeneity must be verified. The Hosking and Wallis' homogeneity tests and the discrepancy test described in the section 2.3 are used. A heterogeneity measure H

of -0.13 was obtained and no discordant sites were found. The region is then considered as statistically homogeneous.

### 3.3.2 The physically and statistically homogeneous region for Brest

In the case of Brest, we note that the region is larger than the one built for Calais (see Fig. 6) with many sites for which the dependency probability is at the limit of the dependency threshold. It is especially the case for sites in the Bristol Channel (UK coast). A study was conducted with and without these sites, which were finally selected in the region. In fact,

their absence led to the selection of a model that did not fit very well with the data. The extremogram between Brest and all other sites is shown in Fig. 5 and Fig. 6 represents the region around Brest. Figure 5 shows, on the vertical axis, the probabilities of the extremal dependence and along the horizontal axis we show sites sorted in ascending order based on the geographical distance to the target site (the closest sites to the target site are on the left). Like for the previous case of Calais, the sites located above the threshold of 0.3 (line in red) are integrated into the region of Brest (considered to be

physically homogeneous). In brackets, next to the name of each site i, durations are indicated for the year in which the tide gauge of the site i and the tide gauge of the target site have operated simultaneously (therefore, it is the number of years on which extremogram was calculated). As shown in Fig. 6, the region around Brest is smaller than that the one which includes Brest in Weiss (2014c) or Weiss et al. (2013) study (see region 1, Fig. 2) but nevertheless is better focused on Brest.

By applying the discrepancy and homogeneity tests, we find no discordant site and the heterogeneity measure *H* is *0.99*; therefore, the region is also considered as statistically homogeneous.

### 3.3.3 The physically and statistically homogeneous region for La Rochelle

The La Rochelle site has been the subject of many studies after the Xynthia storm (e.g. Hamdi et al., 2015). In this study, the region centered on the La Rochelle site raised the question of whether or not to add Saint-Servan and Saint-Malo sites

in the region. Indeed, although Saint-Servan has a dependency extremal probability of *0.4*, this value has been calculated only with a common period with the target site of solely *2* years (which may not be very representative). In addition, Saint-





*Marc Andreewsky*

Malo shares a common period of *14* years with La Rochelle but has an extremal dependency probability of *0.29*, just below the threshold.

St. Malo and St. Servan being very close (with a distance of less than *2* km), it seems logical to either add them both to the region, or withdraw them both to the region. It was finally decided to integrate them into the region of La Rochelle because

Saint Helier (whose abbreviation is "JER" in Fig. 7), very close also to Saint Malo and Saint Servan, was selected to be inside the region of La Rochelle (the dependency extremal probability between Saint Helier and La Rochelle is calculated on *14* years and is equal to *0.3*).

However, these examples show that the extremogram is not a tool to be used blindly and that the choice of the threshold, which serves as an effective aid to form the region as consistently as possible, does not necessarily need to be inflexible.

Figure 7 represents the extremogram for La Rochelle and Fig. 8 represents the region built focused on La Rochelle. As shown in Fig. 8 and as in the case of Calais and Brest, the region around the La Rochelle site is also smaller than that obtained by Weiss et al. (2013), but nevertheless better centered on the La Rochelle site.

By applying the discrepancy test, we find that the site of Eyrac is discordant ($D_c=3.65$). This site is located in the center of the region and this discrepancy could be explained by the specific sea conditions in the Arcachon basin. When Eyrac is

removed from the region of La Rochelle, we find no discordant site and the heterogeneity measure $H$ is *0.53*. So the region (without Eyrac) is also considered statistically homogeneous. The final region for La Rochelle is shown in Fig. 9.

### 3.3.4 Step to build the regional sample

To estimate the regional distribution we used, for each constructed region, the regional pooling method; however, extreme events that can impact several sites (due to the presence of intersite dependence) during a single storm must be considered

only once. Storms elaborate within the procedure described in section 2.2 are a relevant way to suppress the intersite dependence. The distribution of the storm regional maximum denoted $M_s$ is now, for each region, supposed to be identical as the regional distribution. Of course, we must verify the validity of this assumption. In order to evaluate the null hypothesis so that $Z_i$ and $M_s$ have the same distribution for each site $i$, a Kolmogorov-Smirnov test, as explain in section 2.4, can be performed. For the three regions built for Calais, Brest, and La Rochelle, we find that no *p*-values are smaller

than the risk level of *5*%, consequently, for each region, the regional distribution can be estimated from each regional sample $M_s$ (which is considered for each region as the regional sample).

### 3.3.5 Regional effective duration and comparison with results from the study of Weiss (2014c)

The delineated homogeneous region around a target site is characterized (by the nature of the method) by a strong dependence between sites. This spatial dependence impacts, in particular, the regional effective duration (that will be even

lower than if this dependence were high). The effective duration is calculated for each region centered on a target site, as explained in section 2.4, and compared with the effective duration of the region built by Weiss (2014c). Results are shown



in table 1. Remember that Calais is a part of region 2 in the study of Weiss (2014c), and that Brest and La Rochelle are a part of region 1 in the study of Weiss (2014c).

In table 1, one will notice that our effective durations associated with the region of Calais have the same size as those from the region constructed by Weiss (2014c) in which Calais is included.

For the region of Brest and La Rochelle, one will notice that our effective durations are lower than the ones from the region constructed by Weiss (2014c) in which the target site is included. However, in all cases, our durations are higher than those found by a local analysis.

### 3.3.6 Check stationarity

In order to fit a GPD law within a fixed threshold, we have to test the stationarity of our samples. In order to carry out this
for each region, we perform a Student test to check the means' equality of two subsamples of our sample. All tests are completed with a risk level of *5%*.

### 3.3.7 Regional fitting for the region focused on Calais, Brest, and La Rochelle

A Generalized Pareto Distribution (GPD) is fitted to the regional sample, taking into account the four seasons. In Appendix C, details are given about the laws which are used. Eight models are possible, and we must now select, from these models,
the one that best fits our observations. The most commonly used criterion in the literature is the AIC (Akaike Information Criterion), based on the estimate of maximum likelihood. This criterion is used in this study. The Expsin model is selected for Calais and the Gpdcos sin model is selected for Brest and La Rochelle (see Appendix C for the definition of those models). Those models are the same models chosen respectively for regions 2 (which includes Calais) and 1 (which includes Brest and La Rochelle) in the study of Weiss (2014). Figure 10 shows all the fittings which are performed. As we
can see on Fig. 10, the fitting looks good: most of the points are inside the confidence intervals which are not so large. Those elements are also relevant to accept the credibility of the law used for the fitting.

### 3.3.8 Return levels and comparison with return levels obtained from the results of the Weiss (2014c) study procedure

The last step in our regional analysis is to calculate the local quantiles by renormalizing the regional distribution by
multiplying with the local indices. Moreover, by following exactly the same procedure as that followed by Weiss (2014c) to find the quantiles and the 70% of the confidence interval at the 3 sites of interest, we can finally compare results. We then obtain the returns levels shown in table 2. It is worth noting that compared to the procedure followed by Weiss (2014c), the confidence intervals herein are not always larger. In addition, in all cases the 2 studies find confidence intervals with almost the same magnitudes.





*Marc Andreewsky*

**4 Conclusion**

The aim of our study is to achieve extreme statistics on skew storm surges and to reduce uncertainties that are found in a local analysis by using RFA. An important step of the RFA is to form a physically homogeneous region. The method which allows one to shape those physical homogeneous regions is based on the use of the spatial extremogram. It helps to

build the influence region of a target site, and we made the assumption that a relatively high enough probability of dependence allows one to consider the region as physically homogeneous.

In most cases, the approach includes (in the same physically homogeneous region) any site near the target site and we can expect that, most of the time, a target site can't be located at the border between two regions. In the introductory section, it was stated that the method proposed by Weiss (2014c) to form the region of interest leads to the problem of the so-called

"border effect". In our study, the question of "What would be the value of a quantile for a particular site located on the border of a region if this site belonged to the neighboring region?" should not arise often. This is one of the interesting elements of the method we have used. Moreover, the method, unlike the one used by Weiss (2014c), seems to not allow two very distant sites to be in the same region. This is quite comforting because it reduces doubts regarding possible traces of physically heterogeneity that could be generated by relatively distant sites.

In addition, the regions seem more consistent geographically than those obtained by Weiss (2014c) because this method groups sites within a region where the climates are more similar (simply because sites are closer). This leads to a one thousand quantiles and the associated confidence intervals of the same magnitude than those found from the results of Weiss (2014c) study procedure. This observation consolidates the 2 approaches.

Finally, compared to the study of Weiss (2014c), the method used herein can likely increase the level of the physical

homogeneity of the formed regions and decreases, not in all cases but in general, the effective duration of observation. Nevertheless, cases like the site of Calais (which in this study is no longer close to a border and where the effective duration calculated is roughly the same as the one found by Weiss, 2014c) seem to be particularly interesting. A study on Dunkerque, not shown here, leads to the same conclusion as the study carried out for Calais, and we should probably find other sites as interesting as these 2 sites.

Furthermore, physical homogeneity likely has an impact on statistical homogeneity and if we use the criterion of Hosking and Wallis (1997), we can then observe that all regions that we have built are seen as statistically homogeneous. This is unlike those built by Weiss (2014c) where we can find that the regions 1 and 5 are only considered as possibly statistically homogeneous.

However, we can put forward a limitation of the method in the formation of the regions. Indeed, when the tide gauge sites

operated at different times, the common time period between two sites used to calculate the extremogram may be short. Thus, we can sometimes consider an extremogram calculated on a very short series (or even without any data), which may not be relevant. We must therefore consider this detail during the formation of the regions and possibly add or remove a site for which the extremogram does not give us enough relevant information. This disadvantage has been particularly highlighted in a study consisting of forming a region focused on Dieppe (not shown here), where the creation of the region



using the extremogram has not allowed to construct a region roughly focused on Dieppe. This is why it could also be interesting to study the confidence intervals associated with extremograms in order to have more reliability of our estimates of the extremal dependence between sites.

Finally, one of the possible future improvements, would be the considerations of physical data complementary to the use
of the extremogram (such as the atmospheric pressure or the wind speed and direction) that would surely help in the formation of homogeneous regions.

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

**Appendix A: Equations used to calculate the effective duration**

Weiss et al. (2014b) use the following equation to estimate $D_{eff}$:

$$D_{eff} = \varphi \times \sum_{i=1}^{N} \frac{d_i}{N}. \tag{A1}$$

Weiss et al. (2014b) show that $\varphi = \lambda r / \lambda$, where $\lambda_r$ is the average annual number of storms in the region (and $\lambda$ the average number of storms in any site), and, if $n_r$ is the number of storms observed in the region, an estimation of $D_{eff}$ is given by:

$$D_{eff} = \frac{n_r}{\lambda_r}. \tag{A2}$$

Weiss (2014c) use the same equation for $D_{eff}$.

**Appendix B: Data sources and treatment of eustatism**

French data is supplied by SHOM (Service Hydrographique et Océanographique de la Marine, France) and available on the REFMAR (Réseaux de référence des observations marégraphiques) website, while Spanish and UK data are respectively supplied by IEO (Instituto Español de Oceanografía, Spain) and BODC (British Oceanographic Data Centre, UK). The series range from 1846 (Brest, France) to 2011, they show a mean effective duration of 31 years, and they display

missing values.

Local mean sea levels may be affected by eustatism, while tidal predictions are given for the present time. In order to calculate the actual surges of past periods, the sea level must be corrected from a possible eustatism. If annual sea levels, calculated following the PSMSL (Permanent Service for Mean Sea Level) recommendations show significant linear trends, then raw sea level data is corrected to ensure the stationarity of annual sea levels. In regions with strong tidal influence,

coastal flooding hazard is more marked around the times of high tide. Therefore, we restricted our attention to skew surge series, in order to describe the surge contribution at the maximum tidal level. The skew surge is defined as the algebraic difference between the maximum observed sea level around the time of theoretical (predicted) high tide and the predicted high tide level. Thus, the resulting skew surge series have a temporal resolution of approximately 12.4 hours. For a more detailed introduction on skew surges, see, for example, the study of Bernardara et al. (2011).




## Appendix C: Possible laws for the fitting

Let's denote $v$ the regional random variable. A Generalized Pareto Distribution (GPD) is fitted to the regional sample taking into account the four seasons in the following way:

$$\forall v \geq 1, F_r(v) = \sum_{c=1}^{4} p_{r,c} F_{r,c}(v) \tag{C1}$$

$p_{r,c}$ is the frequency of occurrence of season c in the regional sample (empirically estimated as the observed proportion of storms that occurred during season c in the region). We also have:

$$F_{r,c} \sim GPD(1, \gamma_{r,c}, k_r) \tag{C2}$$

and:

$$\log(\gamma_{r,c}) = \gamma_r^0 + \gamma_r^1 \cos\left(\frac{2\pi}{4}c\right) + \gamma_r^2 \sin\left(\frac{2\pi}{4}c\right) \tag{C3}$$

Where $\gamma_r^0, \gamma_r^1, \gamma_r^2$ are *3* parameters to be estimated.

8 models are possible within the possible values of $\gamma_r^0, \gamma_r^1, \gamma_r^2, k_r$ (see Table C1).

## Appendix D: List of ports where tidal measurements are used

List of ports where we have data with their abbreviations (see Table D1).



*Marc Andreewsky*

**Table 1. Comparison of the effective duration found in this study and in the study of Weiss (2014c). The effective duration takes into account the dependence between sites. The total duration is the sum of the observation times of all sites in the region without considering the dependence.**

| Regions | Region focused on Calais | Region focused on Brest | Region focused on La Rochelle | Region 1 from (Weiss, 2014c) | Region 2 from (Weiss, 2014c) |
|---|---|---|---|---|---|
| Effective duration (years) | 151 | 348 | 348 | 517 | 151 |
| Total duration (years) | 375 | 783 | 605 | 1011 | 443 |

**Table 2. Comparison of the one thousand years quantile and the 70% of the confidence interval of this quantile found in this study (RFA Target Focused) and obtained from the results of Weiss (2014c) study procedure, for the three target sites Calais, Brest, and La Rochelle. The confidence intervals are calculated using the bootstrap method like in the study of Weiss (2014c).**

| Estimated values | Type of study | Calais | Brest | La Rochelle |
|---|---|---|---|---|
| Q1000 (m) | RFA Target Focused | 1.55 | 1.68 | 1.68 |
| | Values found from the results of the RFA from Weiss (2014c) study procedure | 1.62 | 1.56 | 1.67 |
| CI70% (m) | RFA Target Focused | 0.18 | 0.19 | 0.17 |
| | Values found from the results of the RFA from Weiss (2014c) study procedure | 0.30 | 0.14 | 0.15 |

**Table C1. Possible models for the fitting of the regional samples.**

| Model Names | Parameters values |
|---|---|
| Exp | $\gamma_r^1 = \gamma_r^2 = k_r = 0, \gamma_r^0 \epsilon R$ |
| Exp$_{cos}$ | $\gamma_r^2 = k_r = 0, (\gamma_r^0, \gamma_r^1) \epsilon R^2$ |
| Exp$_{sin}$ | $\gamma_r^1 = k_r = 0, (\gamma_r^0, \gamma_r^2) \epsilon R^2$ |
| Exp$_{cos\,sin}$ | $k_r = 0, (\gamma_r^0, \gamma_r^1, \gamma_r^2) \epsilon R^3$ |
| Gpd | $\gamma_r^1 = \gamma_r^2 = 0, (\gamma_r^0, k_r) \epsilon R^2$ |
| Gpd$_{cos}$ | $\gamma_r^2 = 0, (\gamma_r^0, \gamma_r^1, k_r) \epsilon R^3$ |
| Gpg$_{sin}$ | $\gamma_r^1 = 0, (\gamma_r^0, \gamma_r^2, k_r) \epsilon R^3$ |
| Gpd$_{cos\,sin}$ | $(\gamma_r^0, \gamma_r^1, \gamma_r^2, k_r) \epsilon R^4$ |

**Table D1. List of ports with their abbreviations.**

| Abbreviations Used | Harbors |
|---|---|



*Extremogram for RFA*

| | |
|---|---|
| ABE | Aberdeen |
| EYRAC | Arcachon |
| AVO | Avonmouth |
| BAN | Bangor |
| BAR | Barmouth |
| BOUCAU | Bayonne |
| BOULOGNE | Boulogne |
| BREST | Brest |
| CALAIS | Calais |
| CHERBOURG | Cherbourg |
| CONCARNEAU | Concarneau |
| CRO | Cromer |
| DEV | Devonport |
| DIEPPE | Dieppe |
| DOV | Dover |
| DUNKERQUE | Dunkerque |
| FEL | Felixstowe |
| FIS | Fishguard |
| HAR | Harwich |
| HEY | Heysham |
| HIN | Hinkley Point |
| HOL | Holyhead |
| ILF | Ilfracombe |
| IMM | Immingham |
| KIN | Kinlochbervie |
| COROGNE | La Coruna |
| LROCH | La Rochelle |
| LCONQ | Le Conquet |
| LCROU | Le Crouesty |
| HAVRE | Le Havre |
| LEI | Leith |
| LER | Lerwick |
| LLA | LIandudno |
| LIV | Liverpool |
| LOW | Lowestoft |
| MHA | Milford Haven |
| MIL | Millport |
| MOY | Moray Firth |
| MUM | Mumbles |
| NHA | Newhaven |
| NEW | Newlyn |
| NPO | Newport |
| NSH | North Shields |
| OLONNE | Olonne |
| IOM | Port Erin |
| PBLOC | Port-Bloc |
| POR | Portpatrick |
| PRU | Portrush |
| PTM | Portsmouth |
| PORT-TUDY | Port-Tudy |
| ROSCOFF | Roscoff |





*Marc Andreewsky*

| | |
|---|---|
| SMALO | Saint Malo |
| SJLUZ | Saint-Jean-de-Luz |
| SANTANDER | Santander |
| SHE | Sheerness |
| JER | St. Helier |
| STM | St. Mary's |
| SNAZA | St-Nazaire |
| STO | Stornoway |
| SERVAN | St-Servan |
| TOB | Tobermory |
| ULL | Ullapool |
| VERDON | Verdon |
| WEY | Weymouth |
| WHI | Whitby |
| WIC | Wick |
| WOR | Workington |

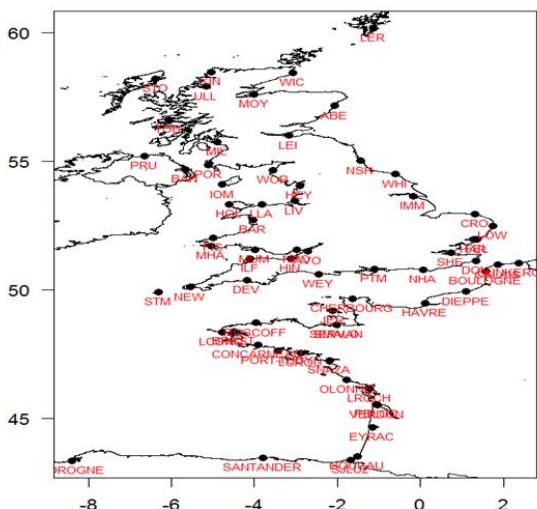

**Fig. 1. The 67 ports along the Spanish, French and U.K. coasts used for the study. Ports are represented by black dots.**




*Extremogram for RFA*

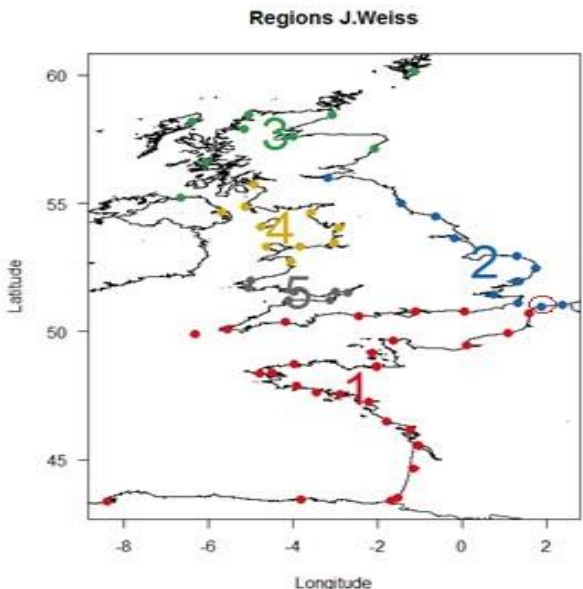

**Fig. 2.** Five physically and statistically homogenous regions within the results of Weiss (2014c) are represented by dots in five colors. Inside the red circle is the site of Calais, very close to a border.

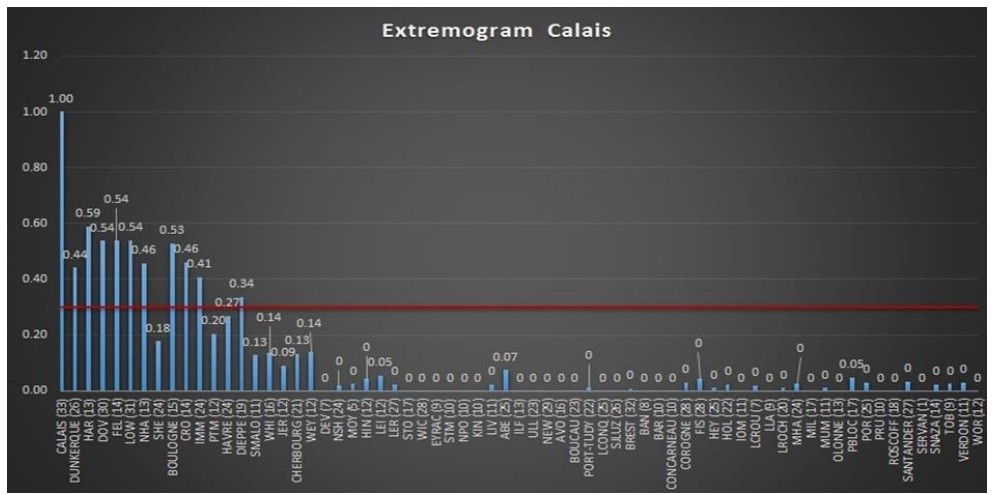

5    **Fig. 3.** Extremogram between Calais and all others sites (see Appendix D for the definition of the abbreviations).





*Marc Andreewsky*

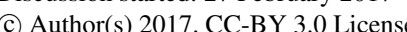

Fig. 4. **Physically homogenous region for Calais represented in red dots found in our study (set number 1). Calais is located inside the red circle. The blue dots (set number 2) represent all the sites which are not a part of the physically homogenous region built around Calais.**

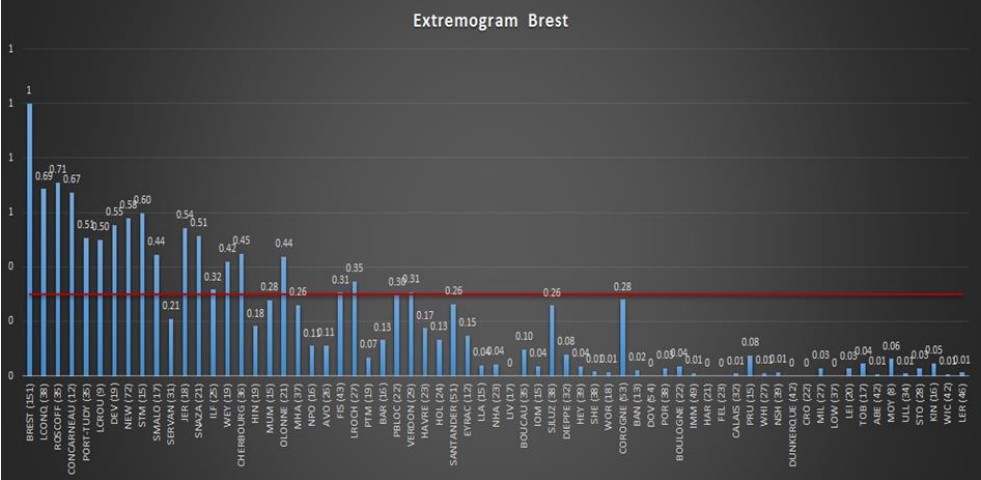

Fig. 5. **Extremogram between Brest and all other sites.**



*Extremogram for RFA*

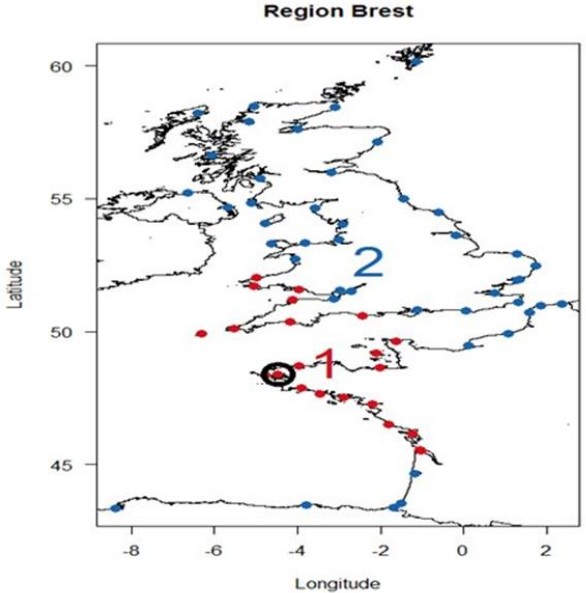

**Fig. 6.** **Physically homogenous region for Brest represented in red dots found in our study (set number 1). Brest is located inside the black circle. The blue dots (set number 2) include all the sites which are not a part of the physically homogenous region built around Brest.**



*Marc Andreewsky*

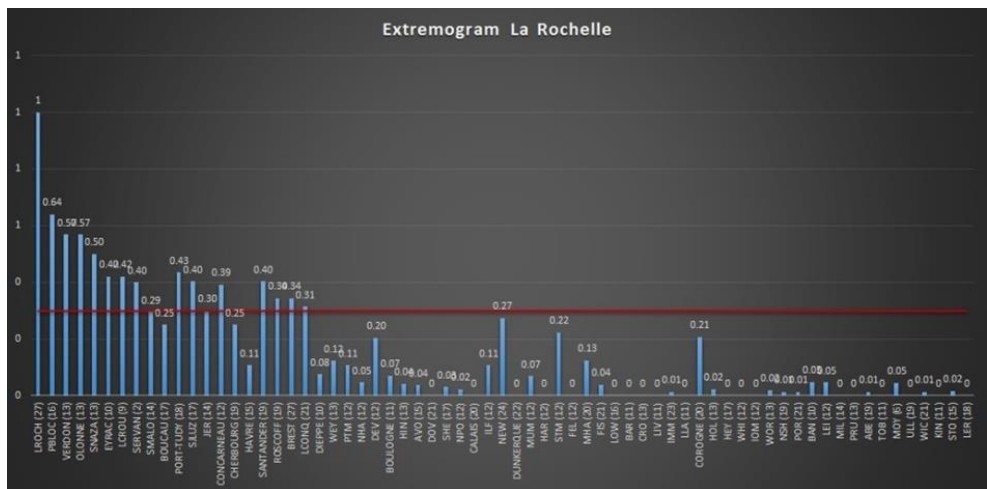

**Fig. 7. Extremogram between La Rochelle and all other sites.**

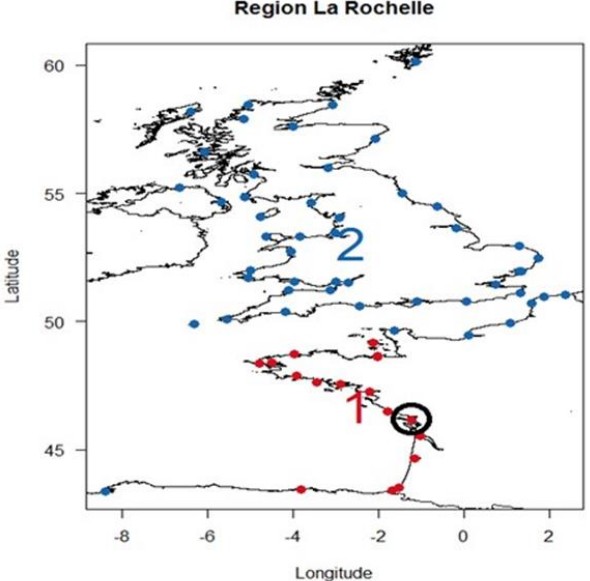

**Fig. 8. Physically homogenous region for La Rochelle represented in red dots found in our study (set number 1). La Rochelle is located inside the black circle. The set number 2 (blue dots) includes all the sites which are not a part of the physically homogenous region built around La Rochelle.**



*Extremogram for RFA*



**Fig. 9. Physically and statistically homogenous region for La Rochelle represented in red dots found in our study (set number 1). La Rochelle is located inside the black circle. The blue dots (set number 2) includes all the sites which are not a part of the physically and statistically homogenous region built around La Rochelle. Eyrac, represented by a blue dot under La Rochelle, is now inside set number 2.**



*Marc Andreewsky*

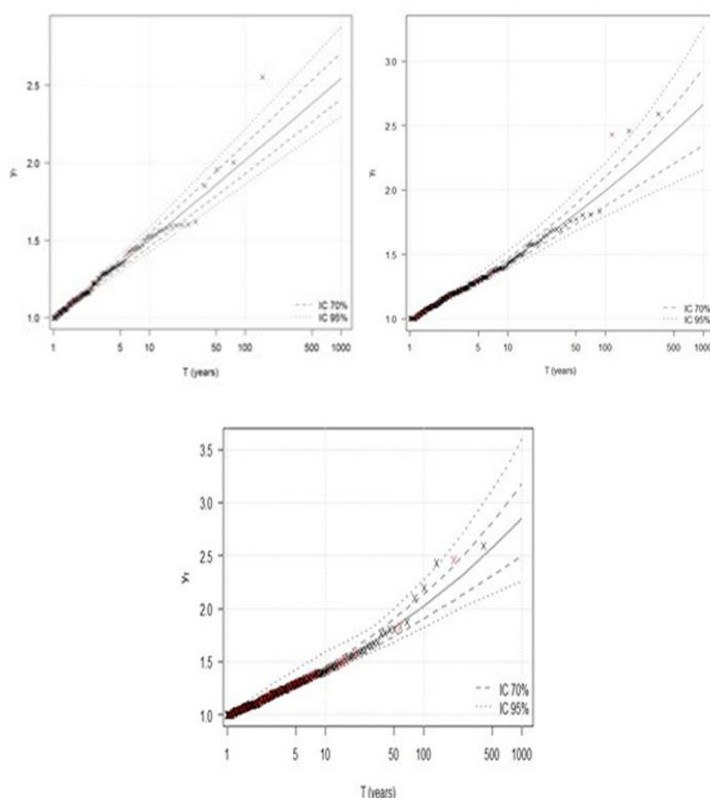

**Fig. 10. Top left, the regional fitting for the region of Calais. Top right, the regional fitting for the region of La Rochelle. On the bottom, the regional fitting for the region of Brest. The red crosses indicate the skew storm surge from the target site, so the black crosses show the contribution to the data of the regional approach. The confidence intervals at 70% and 95% are also shown.**

