# Peer review of "Homogenous regions based on extremogram for regional frequency analysis of extreme skew storm surges"

_Natural Hazards and Earth System Sciences, 2016_

## Referee Comment (RC1) · Anonymous Referee #1 · 7 May 2017

This interesting paper illustrates a methodology for local extreme value analysis using Regional Frequency Analysis (RFA). The technique, given a certain location (let's call it A), identifies a region characterized by the same typology of the extremes and statistically homogeneous, then estimates the return levels of the location of interest as a function of the extremes of the whole area, increasing the sample size and decreasing the uncertainty of the fit. This technique is general, and can be in principle applied to any variable. Here it has been applied by the authors to the skew storm surges.

The identifications of areas of homogeneity of the extremes is an important problem, and this study surely represents an interesting contribution to this topic. I recommend this manuscript for publication after minor revisions.

A limitation I see in this technique is that the extremes hitting locations very close to location A are likely generated by the same events hitting also A, and so add little knowledge. On the other hand, locations far from A (or better, with an extremogram coefficient close to the threshold) are probably hit by many extreme events that do not hit location A, but that are also related with other climates, and therefore it is questionable if we can use them to enlarge the sample of location A.

Anyway I think the advantages of this technique overcome the limitations, also considering that the return levels in specific locations are often very uncertain due to very scant local statistics.

Follows a list of specific comments:

-> In my opinion the word "extremogram" should be defined more clearly. The authors should state that it is a sort of "graph" illustrating the similarity between the extremes of location A and other locations, by means of a measure called extremogram coefficient. I would suggest to give a very synthetic explanation also in the abstract.

-> line 26, pg 1: I would write "... is often too low to obtain accurate estimations of the return levels (associated ..."

-> Eq. 1: the meaning of the superscript -1 is unclear.

-> Eq. 2: this formula is rather unclear to me. What does the "l" in the sums represent? Is this rho the number of peak over threshold occurring in both locations divided by the size of the POT at location A? I would suggest to clarify.

-> line 31, pg 6: lambda == 1: isn't this number of events per year too low? Considering a few events per year would increase the sample, and would be in most cases fitted correctly by a GPD.

-> line 32, pg 6: "is carried out".

-> same line: isn't the KS test too strict? The statistics of the extremes will likely be

slightly different in different locations, due to local conditions, say, of a few percent, much less than the uncertainty in the return levels. But as the sample would increase, the KS test would unavoidably fail beyond whatever risk level, just because the two distributions are slightly different.

-> Figure 1: in the map the names of the locations often overlap. Consider increasing the size of the map, or numbering the locations in the table.

---

## Author Comment (AC1) · 23 May 2017

The authors thank the reviewer for their helpful comments and suggestions. In the text below we have listed the reviewer's comments just after the symbol "->". This is followed by our response. In the supplement, we add the new version of our paper.

-> In my opinion the word "extremogram" should be defined more clearly. The authors should state that it is a sort of "graph" illustrating the similarity between the extremes of location A and other locations, by means of a measure called extremogram coefficient. I would suggest to give a very synthetic explanation also in the abstract.

Yes, that's true that there were ambiguities. This is corrected. A synthetic explanation

is also added in the abstract (line 20, page 1) and in the text (line 2, page 3).

-> line 26, pg 1: I would write "... is often too low to obtain accurate estimations of the return levels (associated ..."

It's corrected.

-> Eq. 1: the meaning of the superscript -1 is unclear.

It's corrected.

-> Eq. 2: this formula is rather unclear to me. What does the "l" in the sums represent? Is this rho the number of peak over threshold occurring in both locations divided by the size of the POT at location A? I would suggest to clarify.

It's corrected: the definition of "l" is added in the text (line 9, page 4).

-> line 31, pg 6: lambda == 1: isn't this number of events per year too low? Considering a few events per year would increase the sample, and would be in most cases fitted correctly by a GPD.

The question is relevant. We have added in the text (line 31, page 6): "Note that, due to the contribution of the others sites in a region, the value of $\lambda$ is not too low to get a lot of events in the regional sample and, in the case of $\lambda=1$, equation A2 shows that the number of the selected skew storm surges is equal to the effective duration. This number, which depends on the region, is high enough (more than 151) in every tested case to carry out the fitting."

-> line 32, pg 6: "is carried out".

It's corrected.

-> same line: isn't the KS test too strict? The statistics of the extremes will likely be slightly different in different locations, due to local conditions, say, of a few percent, much less than the uncertainty in the return levels. But as the sample would increase,

the KS test would unavoidably fail beyond whatever risk level, just because the two distributions are slightly different.

We used the same test as Weiss (2014c) because, one of our goal is to compare our study to Weiss (2014c), and it works: we find that no p-values are smaller than the risk level of 5%. -> Figure 1: in the map the names of the locations often overlap. Consider increasing the size of the map, or numbering the locations in the table.

It's corrected. The new map, clearer, replaces the old map.

Please also note the supplement to this comment:
http://www.nat-hazards-earth-syst-sci-discuss.net/nhess-2016-378/nhess-2016-378-AC1-supplement.pdf

**Supplement:**

**Homogenous regions based on spatial extremogram for regional frequency analysis of extreme skew storm surges**

[revised manuscript text omitted]

---

## Referee Comment (RC2) · Anonymous Referee #2 · 17 Jul 2017

In the context of extreme storm surges, the authors adopt a methodology to define homogeneous regions in which observation follow – apart from a local normalization – the same probability distribution. The idea is that by studying the so-called regional frequency (in contrast to individual sites), uncertainty about the return levels of extreme events can be reduced. The methodology involves a set of statistical steps which are detailed in the manuscript. The authors study a considerable number of sites at the coasts of Great Britain, Atlantic coast of France, and part of the Atlantic coast of Spain. The obtained regions are compared to previous results by Weiss et al. Particular attention is given to the border effects and if so-called target sites are rather not at the border of the obtained regions.

The manuscript is well structured, mostly comprehensive, and seems to be based on a careful analysis. However, in the present form I cannot recommend publication. It is not clear what the added value of the manuscript is. Online the discussion paper is posted as "Review article", but in my point of view it does not extensively "summarize the status of knowledge and outline future directions of research" (see also the rather small number of references). On the other hand, a "Research article" does also not seem to be justified since it does not "report substantial and original scientific results" - at least the addition to the existing literature appears to be minor to me. The authors mostly seem to perform a variation of the analysis previously done by Weiss et al.

Minor comments: - Figures are poor in style and quality - take more care about the paragraphs (ie 1 paragraph, 1 idea) - an alternative approach to the similarity of extreme events could be ROC-curves - what is I in Eq.(2)? - is it feasible to perform an out of sample validation? - auto-correlations can lead to the clustering of extreme events, see "Statistics of return intervals in long-term correlated records" Eichner, PRE, 2007 - sea-level exhibit long-term correlations, see "Evidence for long-term memory in sea level" Dangendorf, GRL, 2014; "Long-term sea level trends: natural or anthropogenic?" Becker, GRL, 2014 - temporal resolution of the data is not clear (different information in main text and appendix) - the usage of GPD "law" is misleading - appendix C is not clear